# Relative Telomere Length and Cardiovascular Risk Factors

**DOI:** 10.3390/biom9050192

**Published:** 2019-05-17

**Authors:** Moritz Koriath, Christian Müller, Norbert Pfeiffer, Stefan Nickels, Manfred Beutel, Irene Schmidtmann, Steffen Rapp, Thomas Münzel, Dirk Westermann, Mahir Karakas, Philipp S. Wild, Karl J. Lackner, Stefan Blankenberg, Tanja Zeller

**Affiliations:** 1Clinic of General and Interventional Cardiology, University Heart Center Hamburg-Eppendorf, 20246 Hamburg, Germany; m.koriath@uke.de (M.K.); Christian_m@gmx.net (C.M.); d.westermann@uke.de (D.W.); m.karakas@uke.de (M.K.); s.blankenberg@uke.de (S.B.); 2DZHK (German Center for Cardiovascular Research), Partner Site Hamburg/Kiel/Lübeck, 20246 Hamburg, Germany; 3Department of Ophthalmology, University Medical Center of the Johannes Gutenberg-University Mainz, 55131 Mainz, Germany; norbert.pfeiffer@unimedizin-mainz.de (N.P.); Stefan.Nickels@unimedizin-mainz.de (S.N.); 4Department of Psychosomatic Medicine and Psychotherapy, University Medical Center of the Johannes Gutenberg-University Mainz, 55131 Mainz, Germany; Manfred.Beutel@unimedizin-mainz.de; 5Institute for Medical Biostatistics, Epidemiology and Informatics, Johannes Gutenberg-University Mainz, 55131 Mainz, Germany; irene.schmidtmann@unimedizin-mainz.de; 6Institute for Molecular Genetics, Johannes Gutenberg University, 55131 Mainz, Germany; Steffen.Rapp@unimedizin-mainz.de; 7DZHK (German Center for Cardiovascular Research), Partner Site Rhine-Main, 55131 Mainz, Germany; tmuenzel@uni-mainz.de (T.M.); philipp.wild@unimedizin-mainz.de (P.S.W.); karl.lackner@unimedizin-mainz.de (K.J.L.); 8Center for Cardiology-Cardiology I, University Medical Center of the Johannes Gutenberg-University Mainz, 55131 Mainz, Germany; 9Center for Translational Vascular Biology (CTVB), University Medical Center of the Johannes Gutenberg-University Mainz, 55131 Mainz, Germany; 10Preventive Cardiology and Preventive Medicine, Center for Cardiology, University Medical Center of the Johannes Gutenberg-University Mainz, 55131 Mainz, Germany; 11Center for Thrombosis and Hemostasis, University Medical Center of the Johannes Gutenberg-University Mainz, 55131 Mainz, Germany; 12Department of Clinical Chemistry and Laboratory Medicine, University Medical Center of the Johannes Gutenberg-University Mainz, 55131 Mainz, Germany

**Keywords:** cardiovascular risk factors, cardiovascular disease, telomeres, telomere length, ageing

## Abstract

(1) Background: Telomeres are repetitive DNA sequences located at the extremities of chromosomes that maintain genetic stability. Telomere biology is relevant to several human disorders and diseases, specifically cardiovascular disease. To better understand the link between cardiovascular disease and telomere length, we studied the effect of relative telomere length (RTL) on cardiovascular risk factors in a large population-based sample. (2) Methods: RTL was measured by a real-time quantitative polymerase chain reaction in subjects of the population-based Gutenberg Health Study (*n* = 4944). We then performed an association study of RTL with known cardiovascular risk factors of smoking status as well as systolic and diastolic blood pressure, body mass index (BMI), LDL cholesterol, HDL cholesterol, and triglycerides. (3) Results: A significant correlation was shown for RTL, with age as a quality control in our study (effect = −0.004, *p* = 3.2 × 10^−47^). Analysis of the relation between RTL and cardiovascular risk factors showed a significant association of RTL in patients who were current smokers (effect = −0.016, *p* = 0.048). No significant associations with RTL were seen for cardiovascular risk factors of LDL cholesterol (*p* = 0.127), HDL cholesterol (*p* = 0.713), triglycerides (*p* = 0.359), smoking (*p* = 0.328), diastolic blood pressure (*p* = 0.615), systolic blood pressure (*p* = 0.949), or BMI (*p* = 0.903). In a subsequent analysis, we calculated the tertiles of RTL. No significant difference across RTL tertiles was detectable for BMI, blood pressure, lipid levels, or smoking status. Finally, we studied the association of RTL and cardiovascular risk factors stratified by tertiles of age. We found a significant association of RTL and LDL cholesterol in the oldest tertile of age (effect = 0.0004, *p* = 0.006). (4) Conclusions: We determined the association of relative telomere length and cardiovascular risk factors in a population setting. An association of telomere length with age, current smoking status, as well as with LDL cholesterol in the oldest tertile of age was found, whereas no associations were observed between telomere length and triglycerides, HDL cholesterol, blood pressure, or BMI.

## 1. Introduction

Telomeres are special structures at the ends of chromosomes. They function as a chromosome-capping mechanism that protects the chromosomes from degradation. Telomeres maintain genomic stability through fusion and prevent cellular senescence or apoptosis. The structure of a telomere comprises repetitive, non-coding DNA sequences and specific protein complexes. Telomere length (TL) decreases with age, and leukocyte telomere length shortening is accelerated by inflammatory responses, oxidative stress, and abnormal cellular senescence [1,2,3,4,5,6]. In patients of advanced age, a correlation of telomere length has been found, amongst others, with cancer [7,8,9], diabetes [10], and cardiovascular disease (CVD) [11,12]. The association of TL with CVD was reported in different studies [11]. Here, a higher relative risk for coronary heart disease as well as cerebrovascular disease was observed in subjects with short leucocyte TL. Accordingly, in study of over 60,000 Danish individuals, shorter TL was associated with an elevated hazard ratio for ischemic heart disease [12].

However, the establishment of a link between TL and CVD, for example, by cardiovascular risk factors, remains unclear. For a better understanding of this link, the relation of TL and cardiovascular risk factors such as hypertension, smoking and lipid levels must be explored [4,13,14,15,16,17,18,19,20,21,22,23].

To broaden the understanding of the relation between TL and cardiovascular risk factors, we studied the effect of relative telomere length (RTL) in leucocytes on cardiovascular risk factors in a large population-based sample.

## 2. Materials and Methods 

### 2.1. Study Population

The study population consisted of 4944 subjects from Gutenberg Health Study (GHS) [24]. GHS is a single-center cohort study with a prospective, observational, and community-based design. Local governmental registry offices in the city of Mainz and the district of Mainz-Bingen were used to randomize the study cohort. Inclusion criteria was age between 35 and 74 years. Individuals with insufficient German language skills as well as physical or psychological inabilities to attend the scheduled study center examinations were excluded. Sample stratification was performed for sex, urban or rural residence, and age. Between 2007 and 2012, 15,010 individuals were included in the study and were seen in the study center for baseline examination. GHS was approved by the Ethics Commission of the State Chamber of Physicians of Rhineland-Palatine (reference no. 837.020.07, original vote March 22, 2007, last update October 20, 2015). Written informed consent was given by all individuals before entering GHS according to the Declaration of Helsinki.

### 2.2. Definition of Cardiovascular Risk Factors

Cardiovascular risk factors and clinical variables were classified on the basis of computer-assisted personal interviews, laboratory examinations, blood pressure, and anthropometric measurements. Standard operation procedures were used for data assessment, which was undertaken by certified medical technical assistants.

The following definitions were used for cardiovascular risk factors. Smoking status was defined as either ever or current smoker based on self-reporting. Anthropometric measurements were performed on calibrated digital scales (Seca 862, Seca, Germany). Body mass index (BMI) was calculated as the body mass in kilograms divided by the square of the body height in meters. Medication to reduce blood pressure, an observed mean systolic blood pressure ≥140 mmHg or an observed mean diastolic blood pressure ≥90 mmHg was classified as hypertension. Blood pressure was measured after 5, 8, and 11 min of rest by an automated sphygmomanometer blood pressure meter (Omron 705CP-II, OMRON Medizintechnik Handelsgesellschaft GmbH, Germany). The second and third standardized measurement was used for the calculation of mean values.

### 2.3. Collection and Processing of Blood Samples

Peripheral blood was collected from each participant of the GHS under fasting conditions and was processed instantly according to standard operating procedures. LDL cholesterol, HDL cholesterol, and triglycerides were measured on the day of sampling by routine methods in the central laboratory of the University Medical Center in Mainz, Germany. Genomic DNA was extracted from peripheral blood leucocytes by extraction from 9 ml EDTA blood samples using the method of Miller [25] and stored at −80 °C until analyses.

### 2.4. Relative Telomere Length Measurements

Relative telomere length was measured from DNA samples extracted from peripheral blood leukocytes by polymerase chain reaction (PCR) according to the method described by Cawthon [26]. The repetitive DNA sequence of telomeres was amplified in combination with the known single copy gene 36B4. Measurements included 5 ng/µL genomic DNA; each PCR plate contained a quality check, non-template controls, and reference DNA aliquots. Reference DNA was used as a calibrator. Dilution of DNA samples and pipetting of diluted DNA and PCR master mix into PCR plates were performed using an automated liquid handling system (Bravo Automated Liquid Handling Platform, Agilent, USA). The experiments were performed in two batches. Experiments in batch one were performed first and were performed in duplicate on two different PCR plates. Experiments in batch two were performed in triplicate on the same PCR plate. Raw data were analyzed using linregPCR (version 2012.0, 2011, no publisher), a software designed to calculate PCR efficiencies and standardized Ct values [27,28]. To calculate the relative telomere length (RTL) for each DNA sample, the relative telomere to single copy gene (T/S) ratio was calculated. T/S describes the factor by which the sample differed from a reference DNA sample in its ratio of telomere repeat copy number to single copy gene copy number. The RTL was proportional to the average telomere length [26] and thus was used for further analyses. The measurements were performed on an Applied Biosystems Taqman Fast Real-Time PCR 7900 HT System (Applied Biosystems, Darmstadt, Germany). Valid RTL measurements were available for 4080 subjects.

### 2.5. Statistical Analysis

Statistical analyses were performed using R (R Development Core Team, version 3.4.3, www.r-project.org/). A cutoff of a RTL > 3 or < 0.33 was arbitrarily chosen to identify and remove outliers. Two additional steps, based on deviation from the mean, were applied to remove outliers: (i) PCR plate-wise exclusion of samples deviating > 3 sd from the mean RTL and (ii) Overall exclusion of samples deviating > 3sd from the mean RTL. The workflow of our quality control is shown in Appendix A.

RTL was adjusted for age to consider the known negative correlation of RTL with age [29,30]. Adjusted RTL was then used in a multivariable linear regression analysis to assess the relation of RTL to both age and cardiovascular risk factors. PCR plate and batch as possible confounders were accounted for during the analysis. The known relation of RTL and age was used as a quality control. The population was divided into three tertiles depending on their RTL. The tertiles with the longest RTL was used as a reference and compared to the other tertiles in a multivariable linear regression analysis for quantitative traits. For additional analysis, samples were divided into tertiles of age. Multivariable linear regression analysis for RTL with cardiovascular risk factors was then calculated for each tertile of age separately.

Associations with a *p*-value ≤ 0.05 were considered significant.

## 3. Results

4,080 samples had valid RTL measurements and were available for statistical analysis. Characteristics of the study population are shown in Table 1. Out of the subjects, 48.9 percent were female. Mean body mass index (BMI) was 27.2 (standard deviation [SD]: 4.7) kg/m^2^, mean systolic blood pressure was 133 (SD: 18) mmHg, mean diastolic blood pressure was 83 (SD: 10) mmHg, mean LDL cholesterol was 140.0 (SD: 36.2) mg/dL, mean HDL cholesterol was 56.5 (SD: 15.8) mg/dL, and mean triglycerides were 126.2 (SD: 74.4) mg/dL.

A significant correlation was shown for RTL with age as a quality control in our study (effect = −0.004, 95% confidence interval (CI) (−0.005–−0.004), *p* = 3.2 × 10^−47^) (Figure 1). Analysis of the relation between RTL and cardiovascular risk factors showed a significant association of RTL with current smoking (effect = −0.016, CI (−0.031–0.000), *p* = 0.048) (Table 2). After correction for multiple testing, the effect did not remain significant (false discovery rate, FDR = 0.383). No significant associations with RTL were seen for the other cardiovascular risk factors, namely LDL cholesterol (*p* = 0.127), HDL cholesterol (*p* = 0.713), triglycerides (*p* = 0.359), ever smoking (*p* = 0.328), diastolic blood pressure (*p* = 0.615), systolic blood pressure (*p* = 0.949), and BMI (*p* = 0.903).

Subsequently, we calculated tertiles of the RTL and compared RTL tertiles for the quantitative cardiovascular risk factors BMI, systolic and diastolic blood pressure, LDL cholesterol, HDL cholesterol, trigylcerides, and smoking status. No significant difference across RTL tertiles was detectable (Figure 2). Also, current and ever smoking showed no significant difference across RTL tertiles (*p* = 0.072 for current smoking and *p* = 0.156 for ever smoking).

We also studied the association of RTL with cardiovascular risk factors according to tertiles of age. Subjects in tertile 1 (*n* = 1437) were between 35 and 50 years of age, in tertile 2 (*n* = 1336) between 51 and 62 years, and in tertile 3 (*n* = 1298) between 63 and 74 years. For subjects in the oldest tertile of age, a significant correlation of RTL with LDL cholesterol was found (effect = 0.0004, CI (0.000–0.001), *p* = 0.006) (Table 3). An association of RTL with LDL cholesterol was not seen in the other two tertiles of age (tertile 1 *p* = 0.421, tertile 2 *p* = 0.900). There were no significant association of RTL with other cardiovascular risk factors: BMI, systolic and diastolic blood pressure, HDL cholesterol, triglycerides, and smoking when stratified by tertiles of age.

## 4. Discussion

To gain a better understanding of the link between telomere length (TL) and cardiovascular disease (CVD), we conducted an association study of relative telomere length (RTL) with cardiovascular risk factors of BMI, blood pressure, lipid levels, and smoking status in a large sample of the general population. To our knowledge, no association study for cardiovascular risk factors of this size has been undertaken in the German population to date.

Our primary results showed (i) a strong correlation of RTL with age, (ii) a modest association of RTL with current smoking status and no significant associations of RTL with lipid levels, blood pressure or BMI, (iii) no significant association of RTL with cardiovascular risk factors when analyzing tertiles of RTL, and (iv) a modest association of RTL with LDL cholesterol in the oldest tertile of age.

There have been conflicting results in smaller sample size studies regarding the relation of TL and cardiovascular risk factors [4,13,14,16,18,19,22,23]. Some studies have reported similar results, including an association of TL and smoking in women [17] and older men [21] but not in blood donors [15]. The relation of smoking and TL was also analyzed in a large-scale population based study by the National Health and Nutrition Examination Surveys for 1999 to 2002 (NHANES) [31]. In this study, no significant association of TL and smoking was found. However, the NHANES data set was extensively analyzed with different scopes by several study groups. Results involving TL and cardiovascular risk factors were conflicting, suggesting that the reported associations might not to be robustly reproducible [32,33,34,35].

An association of smoking and TL was also shown in a large-scale study in the Danish population [36]. However, in comparison with our study, the analyses in this Danish study found associations among TL and total cholesterol, triglycerides, and BMI [36]. A possible explanation of these differences relate to the methodological nature of the study, as the Danish study did not analyze TL as a continuous variable as we did, indicating the need to keep statistical methods in mind when comparing differing study results.

The modest association of RTL with current smoking status which we found in our initial analysis could neither be replicated in our analysis of RTL tertiles nor in our age-stratified analysis. This is possibly due to reduced statistical power when dividing the data into tertiles.

Taken together, our data provides evidence that the known association of telomere length and CVD is possibly not mediated through a strong effect of telomere length on cardiovascular risk factors.

Some limitations of our study merit consideration. We were not able to detect possible effects that might evolve over time because no longitudinal data on RTL are available. Additionally, the study population used in this study had a maximum age limit of 74 years of age. Older subjects might indicate a stronger effect of TL on cardiovascular risk factors than did our study population. Our additional finding of a modest association of RTL with LDL cholesterol in the oldest tertile of age points in this direction. Furthermore, automated assays might reduce variability of TL measurements. However, such assays were not available when we started our study. Finally, a causative effect of RTL and CVD could not be investigated because of the observational design of our study.

## 5. Conclusions

In a population setting, we determined the association between relative telomere length and cardiovascular risk factors. An association of RTL with age, current smoking status, as well as with LDL cholesterol in the oldest tertile of age was found; in contrast, no associations were observed between RTL and triglycerides, HDL cholesterol, blood pressure, or BMI. The results suggest that the known association of telomere length and CVD is not likely to be mediated through a strong effect of telomere length on cardiovascular risk factors.

## Figures and Tables

**Figure 1 biomolecules-09-00192-f001:**
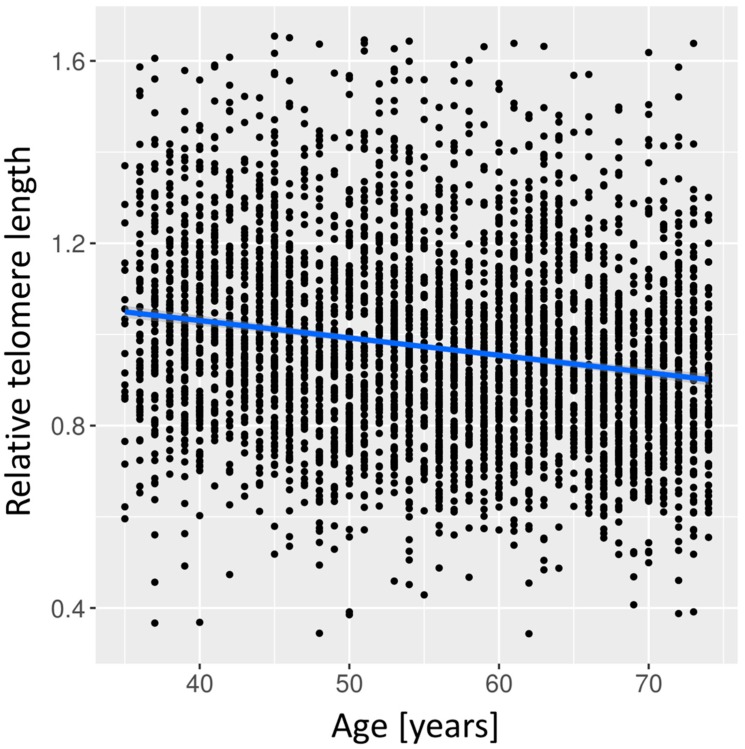
Correlation of relative telomere length with age. Relative telomere length was adjusted for age, batch, and PCR plate. Effect = −0.004, 95% confidence interval (−0.005–−0.004), *p*-value = 3.2 × 10^−47^.

**Figure 2 biomolecules-09-00192-f002:**
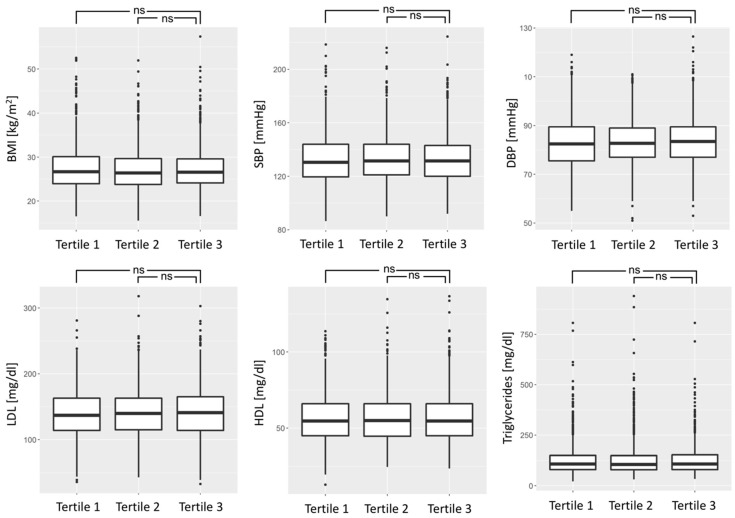
Relative telomere length tertiles for cardiovascular risk factors Data shown for body mass index (BMI), systolic blood pressure (SBP), diastolic blood pressure (DBP), LDL cholesterol (LDL), HDL cholesterol (HDL), and triglycerides. Tertile 1 represents shortest RTL. Differences were statistically non-significant (ns).

**Table 1 biomolecules-09-00192-t001:** Baseline characteristics of the population studied. Continuous variables are shown as mean values (standard deviation), and categorical variables are shown as numbers (%).

N	4080
Age, years	55.7 (10.9)
Females	1997 (48.9)
BMI, kg/m^2^	27.2 (4.7)
Systolic blood pressure, mmHg	132.9 (17.7)
Diastolic blood pressure, mmHg	83.2 (9.5)
LDL-cholesterol, mg/dL	140.0 (36.2)
HDL-cholesterol, mg/dL	56.5 (15.8)
Triglycerides, mg/dL	126.2 (74.4)
Ever smoker	2148 (53.3)
Current smoker	764 (18.8)

**Table 2 biomolecules-09-00192-t002:** Multivariable linear regression analysis of relative telomere length on cardiovascular risk factors. Relative telomere length was adjusted for age, batch, and PCR plate.

Cardiovascular Risk Factor	Effect	95% Confidence Interval	P-Value	False Discovery Rate
Current smoker	−0.015	−0.031–0.000	0.048	0.383
Ever smoker	−0.006	−0.018–0.006	0.328	0.718
Triglycerides	0.000	0.000–0.000	0.359	0.718
LDL-cholesterol, mg/dL	0.000	0.000–0.000	0.127	0.506
HDL-cholesterol, mg/dL	−0.000	−0.001–0.000	0.713	0.949
BMI, kg/m^2^	0.000	−0.001–0.001	0.903	0.949
Systolic blood pressure, mmHg	0.000	0.000–0.000	0.949	0.949
Diastolic blood pressure, mmHg	0.000	−0.001–0.001	0.615	0.949

**Table 3 biomolecules-09-00192-t003:** Multivariable linear regression of relative telomere length on cardiovascular risk factors according to tertiles of age. Relative telomere length was adjusted for age, batch, and PCR plate.

	Tertile 1 (35 years–50 years) *n* = 1437				Tertile 2 (51 years–62 years) *n* = 1336				Tertile 3 (63 years–74 years) *n* = 1298			
Cardiovascular Risk Factor	Effect	95% Confidence Intervall	P-Value	False discovery RATE	Effect	95% Confidence Intervall	P-value	False Discovery Rate	Effect	95% Confidence Intervall	P-Value	False Discovery Rate
Current smoker	−0.019	−0.042–0.004	0.103	0.517	−0.021	−0.047–0.005	0.120	0.371	0.006	−0.030–0.042	0.752	0.772
Ever smoker	−0.006	−0.026–0.015	0.603	0.799	−0.013	−0.034–0.009	0.260	0.371	0.005	−0.016–0.027	0.617	0.772
Triglycerides	0.000	0.000–0.000	0.058	0.517	0.000	0.000–0.000	0.106	0.371	0.000	0.000–0.000	0.323	0.656
LDL-cholesterol, mg/dL	0.000	0.000–0.000	0.421	0.799	0.000	0.000–0.000	0.898	0.898	0.0004	0.000–0.001	0.006	0.058
HDL-cholesterol, mg/dL	0.001	0.000–0.001	0.186	0.619	−0.001	−0.001–0.000	0.205	0.371	0.000	−0.001–0.001	0.638	0.772
BMI, kg/m^2^	−0.001	−0.003–0.002	0.573	0.799	0.002	−0.001–0.004	0.165	0.371	−0.002	−0.004–0.001	0.154	0.512
Systolic blood pressure, mmHg	0.000	−0.001–0.001	0.880	0.880	0.000	−0.001–0.001	0.558	0.697	0.000	−0.001–0.000	0.394	0.656
Diastolic blood pressure, mmHg	0.000	−0.001–0.002	0.639	0.799	0.000	−0.001–0.001	0.843	0.898	0.000	−0.001–0.001	0.772	0.772

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
