# Peer review of "Relative Telomere Length and Cardiovascular Risk Factors"

_biomolecules, 2019, doi:10.3390/biom9050192_

Round 1
Reviewer 1 Report
For decades, there are speculations that telomeres and CVD (and metabolic risk factors) are associated. Current efforts meta-analyzing these observatinal studies observed mostly consistent results. The present study also aims to better understand the link between cardiovascular disease and telomere length in a large population-based sample. RTL was measured by real-time quantitative polymerase chain reaction in subjects of the population-based Gutenberg Health Study. The authors performed correlation analyses of RTL with cardiovascular risk factors smoking status, systolic and diastolic blood pressure, BMI, LDL-cholesterol, HDL-cholesterol and triglycerides. And they found significant correlations for RTL with age and smoking.
The study added further evidence in the TL and CVD field.
Tha authors are advised to perform multivariable regression analysis, taking into account of potential confounders. The effect sizes should be reported with 95% confidence intervals.
Author Response
1.) The authors are advised to perform multivariable regression analysis, taking into account of potential confounders.
We agree with the reviewer and apologize for not having been clear enough in our methods section. Relative telomere length (RTL) was adjusted for age and was then taken forward to a multivariable regression analysis with cardiovascular risk factors. PCR plate and batch of the samples as potential confounders were thus accounted for.
The following paragraph in section 2.5 on page 3-4 was rewritten:
RTL was adjusted for age to consider the known negative correlation of RTL with age [29,30]. Adjusted RTL was then used in a multivariable linear regression analysis to assess the relation of RTL and age and cardiovascular risk factors. PCR plate and batch as possible confounders were accounted for in the analysis. The known relation of RTL and age was used as a quality control. For further analysis, the population was divided into three tertiles depending on their RTL. The tertiles with the longest RTL was used as a reference and compared to the other tertiles in a multivariable linear regression analysis for quantitative traits. For another additional analysis, samples were divided into tertiles of age. Multivariable linear regression analysis for RTL with cardiovascular risk factors was then calculated for each tertile of age separately.
2.) The effect sizes should be reported with 95% confidence intervals.
To report our results in a more complete way, we thank the reviewer for this comment and included 95% confidence intervals in the results.
The following two sentences were changed in section 3 on page 4:
A significant correlation was shown for RTL with age as a quality control in our study (effect=-0.004, 95% confidence intervall (CI) (-0.005 - -0.004), p=3.2x10-47) (Figure 1). Analyzing the relation of RTL and cardiovascular risk factors, a significant association of RTL with current smoking was found (effect=-0.016, CI (-0.031-0.000), p=0.048) (Table 2).
95% confidence intervalls were included in the caption of figure 1 in section 3 on page 5:
Figure 1. Correlation of relative telomere length with age Relative telomere length was adjusted for age, batch, and PCR plate. Effect=-0.004, 95% confidence intervall (-0.005 - -0.004), p-value=3.2x10-47.
Table 2 in our manuscript on page 5 was changed accordingly:
Cardiovascular risk factor | Effect | 95% confidence intervall | P-value | False discovery rate |
Current smoker | -0.015 | -0.031-0.000 | 0.048 | 0.383 |
Ever smoker | -0.006 | -0.018-0.006 | 0.328 | 0.718 |
Triglycerides | 0.000 | 0.000-0.000 | 0.359 | 0.718 |
LDL-cholesterol, mg/dl | 0.000 | 0.000-0.000 | 0.127 | 0.506 |
HDL-cholesterol, mg/dl | -0.000 | -0.001-0.000 | 0.713 | 0.949 |
BMI, kg/m2 | 0.000 | -0.001-0.001 | 0.903 | 0.949 |
Systolic blood pressure, mmHg | 0.000 | 0.000-0.000 | 0.949 | 0.949 |
Diastolic blood pressure, mmHg | 0.000 | -0.001-0.001 | 0.615 | 0.949 |
Table 2. Multivariable linear regression analysis of relative telomere length on cardiovascular risk factors Relative telomere length was adjusted for age, batch, and PCR plate.

Reviewer 2 Report
Methods: Please clarify standardization procedure of Ct values in more in detail. Did the authors use a calibrator to calculate Ct values of T and S?
In the present study, the only association found by authors, although very weak (p=0.048), is between RTL and current smoking. Did the authors evaluate associations of RTL with cardiovascular risk factor according to tertiles of age?
Major limit of the study seems to be the non-automated procedure, which insert possible methodological uncertainty in the RTL measurement on a large population
Author Response
1.) Methods: Please clarify standardization procedure of Ct values in more in detail. Did the authors use a calibrator to calculate Ct values of T and S?
RTL was calculated using the formula:
Ct2 are reference DNA values and Ct1 are sample DNA values for the two different primer pairs. The reference DNA was used as calibrator. PCR efficiencies and Ct values were taken from the analysis of the raw data with LinRegPCR. LinRegPCR is a computer program that uses linear regression on the measured fluorescence per PCR cycle to achieve a common window of linearity. Common window of linearity is then used to calculate starting concentrations, PCR efficiencies and standardized Ct values. Mean PCR efficiencies per primer pair per plate were used in our analysis.
The following sentence was rephrased in section 2.4. on page 3:
Measurements included 5ng/µl genomic DNA and each qPCR plate contained a quality check, non-template controls and reference DNA aliquots.
The following sentence was included in section 2.4 on page 3:
Reference DNA was used as a calibrator.
2.) In the present study, the only association found by authors, although very weak (p=0.048), is between RTL and current smoking. Did the authors evaluate associations of RTL with cardiovascular risk factor according to tertiles of age?
We thank the reviewer for his comment and analyzed the association of RTL with cardiovascular risk factors according to tertiles of age as suggested.
The following sentence was included in the abstract on page 2:
Finally, we studied the association of RTL and cardiovascular risk factors stratified by tertiles of age. We found a significant association of RTL and LDL-cholesterol in the oldest tertile of age (effect=0.0004, p=0.006).
The following sentence was adapted to the new finding in the abstract on page 2:
An association of telomere length with age, current smoking, as well as with LDL-cholesterol in the oldest tertile of age was found, whereas no associations were observed between telomere length and triglycerides, HDL-cholesterol, blood pressure or BMI.
The following sentence was included in section 2.5 on page 4:
For another additional analysis, samples were divided into tertiles of age. Multivariable linear regression analysis for RTL with cardiovascular risk factors was then calculated for each tertile of age separately.
The following paragraph and table 3 were included in section 3 on page 6 and 7:
For further analysis we studied the association of RTL with cardiovascular risk factors accoding to tertiles of age. Subjects in tertile 1 (n=1437) were aged between 35 and 50 years, in tertile 2 (n=1336) between 51 and 62 years, and in tertile 3 (n=1298) between 63 and 74 years. For subjects in the oldest tertile of age a significant correlation of RTL with LDL-cholesterol was found (effect=0.0004, CI (0.000-0.001), p=0.006) (table 3). An association of RTL with LDL-cholesterol was not seen in the other two tertiles of age (tertile 1 p=0.421, tertile 2 p=0.900). There were no significant association of RTL with the other cardiovascular risk factors BMI, systolic and diastolic blood pressure, HDL-cholesterol, triglycerides and smoking stratified by tertiles of age

Round 2
Reviewer 2 Report
Overall, the paper in the present form is clearer and more complete and the new analysis added more streght to results and conclusions.